# Mesenchymal–Epithelial Transition Kinase Inhibitor Therapy in Patients with Advanced Papillary Renal-Cell Carcinoma: A Systematic Review and Meta-Analysis

**DOI:** 10.3390/ijms242417582

**Published:** 2023-12-18

**Authors:** Francisco Cezar Aquino de Moraes, Maysa Vilbert, Vinícius Freire Costa Alves, Gustavo de Oliveira Almeida, Jonathan N. Priantti, Thiago Madeira, Carlos Stecca, Marianne Rodrigues Fernandes, Ney Pereira Carneiro dos Santos

**Affiliations:** 1Oncology Research Center, Federal University of Pará, Belem 66075-110, Brazil; 2Department of Medical Oncology and Hematology, Princess Margaret Cancer Centre, University Health Network, Toronto, ON M5T 2S8, Canada; 3School of Medicine, University of São Paulo—USP, São Paulo 14040-904, Brazil; 4School of Medicine, Federal University of Triângulo Mineiro—UFTM, Uberaba 38280-000, Brazil; 5School of Medicine, Federal University of Amazonas—UFAM, Manaus 69020-160, Brazil; 6School of Medicine, Federal University of Minas Gerais—UFMG, Belo Horizonte 31270-901, Brazil; 7Mackenzie Evangelical University Hospital, Curitiba 80710-390, Brazil

**Keywords:** MET status, MET inhibitors, advanced papillary renal-cell carcinoma

## Abstract

Papillary subtypes of renal-cell carcinoma (pRCC) represent 10–15% of the cases and commonly have MET alterations. This systematic review and single-arm meta-analysis evaluated MET inhibitor therapy (METi) efficacy and safety in adults with confirmed advanced pRCC. The search strategy included PubMed, Web-of-science, Cochrane, and Scopus. We used the DerSimonian/Laird random effect model for all analyses; *p*-value < 5% was considered significant, and heterogeneity was assessed with I^2^. Three clinical trials and six cohort studies were included with 504 patients; 31% were MET-driven. Our pooled analysis demonstrated an objective response rate (ORR) in MET-driven, MET-independent, and overall patients of: 36% (95%CI: 10–62), 0% (95%CI: 0–3), and 21% (95%CI: 1–41), respectively. One-year disease control and progression-free survival rates were, respectively, 70% (95%CI: 52–88) and 15% (95%CI: 10–20). Twelve- and twenty-four-month survival rates were, respectively, 43% (95%CI: 23–64) and 10% (95%CI: 0–30). The prevalence of adverse events of any grade and grades 3–5 were 96% (95%CI: 91–100) and 44% (95%CI: 37–50), respectively. We suggest METi has anti-tumor activity and is tolerable in patients with advanced pRCC.

## 1. Introduction

While renal-cell carcinoma (RCC) represents only 2% of cancer diagnoses and fatalities globally, its incidence has doubled in developed nations over the past half century. As a result, it now ranks as the ninth most common neoplasm in the United States (US) [1]. In 2023, kidney and renal pelvis neoplasms are estimated to account for 81,800 new cases and 14,890 deaths in the US [2]. Smoking, obesity, hypertension, chronic renal failure, and dialysis are some well-known risk factors that increases the risk of developing RCC [3,4,5,6]. Additionally, chemical products such as petroleum, asbestos, cadmium, benzene, vinyl chloride, herbicides, and paracetamol abuse can increase the risk of this type of neoplasm [7,8].

Papillary renal cell carcinoma (pRCC), comprising 10–15% of all RCC cases, ranks as the second most common subtype after clear-cell RCC (ccRCC) [9]. The International Metastatic RCC Database Consortium (IMDC) study revealed that pRCC exhibits inferior survival outcomes when compared to metastatic ccRCC, irrespective of metastatic site [10]. PRCC is more frequent in men, occurring mainly between ages 50 and 79 [10,11], and black individuals face a threefold higher risk of developing this type of cancer compared to whites [12].

On imaging, pRCC present as calcifications with a frequent multifocal nature. Histologically, the presence of basophilic or eosinophilic cells in a papillary or tubular architecture distinguishes pRCC from ccRCC [9]. There are two types of pRCC: Type 1, identified by immunoreactivity for CK7 and AMACR [13,14], typically features well-circumscribed renal cortices with papillary and tubular architecture. It includes a layer of cuboidal cells with a low-grade nucleus, often accompanied by foamy histiocytes. Type 2 pRCC has a higher nuclear grade and mitotic index, with several layers of eosinophilic cells. It is currently recognized that type 1 and type 2 pRCC often co-exist [15]. Type 1 commonly shows gains in chromosomes 7 and 17, while type 2 exhibits gains in chromosomes 12, 16, and 20 [16,17], along with gains in chromosomes 2 and 3 [18]. Regarding molecular characteristics, type 1 tumors exhibit alterations in MET status in 81% of cases, with significantly higher MET mRNA expression levels compared to type 2 tumors [9]. Amplification, mutation, and fusion of MET play crucial roles in driving oncogenic processes [19,20].

Given that MET is a recognized abnormality in the pRCC, it represents a promising target for precision therapies. Numerous endeavors have focused on developing MET inhibitors for application in various solid tumors, including pRCC [21]. There are two primary methods for inhibiting the HGF/c-MET axis. The first involves employing monoclonal antibodies (mAbs) to disrupt HGF/c-MET interaction and/or c-MET dimerization. The second method consists of utilizing small molecule c-MET kinase inhibitors, which target activation sites within the receptor’s cytoplasmic domain, preventing its phosphorylation and downstream signaling cascade [22,23].

Treatment with new options that act on alterations in oxygen metabolism and the cell cycle has revolutionized the treatment of RCC [24]. At the beginning of the 21st century, therapies focused on the Von Hippel–Lindau/hypoxia-inducible factor (VHL/HIF) pathway, such as vascular endothelial growth factor (VEGF), transforming growth factor α (TGFα), and platelet-derived growth factor β (PDGFβ) [25,26]. These therapies have shown significant anti-tumor activity for patients with RCC, a tumor considered resistant to chemotherapy [27].

In recent decades, immune checkpoint inhibitors (ICIs) have changed the treatment paradigm for RCC [28,29]. However, most randomized clinical trials with ICIs include patients with clear-cell histology [30]. Only a third of RCCs with other forms of histology respond to ICIs [31,32]. Concerning target therapy strategies, tyrosine kinase inhibitors (TKIs) and mammalian target of rapamycin (mTOR) inhibitors currently have limited data and should be investigated in patients with histology other than ccRCC [33].

Although agents targeting the c-MET pathway showed promise against pRCC, the available data remains limited, rendering this approach experimental. Therefore, our study aims to investigate the anti-tumor activity of c-MET inhibitor therapy by analyzing efficacy outcomes, including progression-free survival (PFS), overall survival (OS), overall response rate (ORR), disease control rate (DCR), and adverse events (AEs), in patients with advanced or metastatic pRCC.

## 2. Materials and Methods

### 2.1. Protocol and Registration

Our meta-analysis followed the guidelines of the declaration Preferred Reporting Items for Systematic Reviews and Meta-Analysis (PRISMA) [34] and the recommendations of the Cochrane Collaboration. This review was registered on the Prospective International Registry of Systematic Reviews—PROSPERO (http://www.crd.york.ac.uk/ (accessed on 9 April 2023)) under the number CRD42023412668.

### 2.2. Eligibility Criteria

We included in this review adult patients (≥18 years) with confirmed papillary renal-cell carcinoma (pRCC) who have not used MET inhibitor treatment for at least six months, excluding vascular endothelial growth factor-directed agents. Patients with MET-driven pRCC—defined as any of the following: chromosome 7 copy gain, focal MET or HGF gene amplification, or MET kinase domain mutations, including or not amplification of the cited gene—were considered eligible. MET kinase inhibitor treatments included foretinib, cabozantinib, crizotinib, savolitinib, and tivantinib. 

Case reports, reviews, opinion articles, technical articles, guidelines, animal studies, and in vitro experiments were excluded. In addition, only articles in English were included, with no restrictions on the publication date of the included articles.

### 2.3. Search Strategy

We systematically searched Pubmed, Cochrane Central, Scopus, and Web of Science for studies published in English with the Medical Subject Headings (MeSH) terms carcinoma, renal cell, and crizotinib; and text words related to kidney neoplasms; papillary renal-cell carcinoma; protein kinase inhibitors; proto-oncogene proteins c-MET; cabozantinib; savolitinib; foretinib; tivantinib; advanced; metastatic; locally advanced; unresectable; and bilateral multifocal. In addition, research was also carried out in abstracts, articles, and scientific presentations of the American Society of Clinical Oncology (http://www.asco.org/ASCO (accessed 27 April 2023)) and the European Society for Medical Oncology (https://www.esmo.org/ (accessed on 27 April 2023)).

Both the MeSH and the input terms underwent adaptations for the selected databases, combining terms with Boolean connectors (OR, AND) and obeying the syntax rules in each base, as described in Appendix A. We screened the references of the included articles and systematic reviews for additional studies. Also, an alert was established for notifications in each database in case a study corresponding to our search was eventually published. We used the reference manager software EndNote^®^ (version X7, Thomson Reuters, Philadelphia, PA, USA). The duplicate articles were automatically and manually excluded. Titles and abstracts of articles found in the databases were analyzed independently by two reviewers (VA and GA). In case of discrepancy between reviewers, a third reviewer was responsible for the final decision for inclusion (FM).

### 2.4. Data Extraction and Risk of Bias Assessment

To summarize the main findings, we collected the following data from the included articles: authors and year, study design, patient sample characteristics (age, race, sex, clinical and pathological data, and study group), and outcomes such as overall survival (OS), progression-free survival (PFS), therapy safety, intervention-related adverse events, duration of response, and objective response rate (ORR). We considered the Disease Control Rate (DCR) as a summation of complete response (CR), partial response (PR), and stable disease (SD), while the objective response rate (ORR) included patients with CR and PR. Toxicities were measured according to the Common Terminology Criteria for Adverse Events (CTCAE, version 5.0) and tumoral response using the Radiological Criteria for the Evaluation of the Oncological Response of Solid Tumors (RECIST, version 1.1).

Two researchers (FM and VA) independently analyzed the risk of bias. In case of disagreement, a third reviewer was responsible for the final decision (MV). Risk of bias assessment was made through Cochrane RoB2 and ROBINS-I tools, respectively, for randomized controlled trials and observational studies [35,36]. Appendix A summarize the risks of bias for the studies selected in this meta-analysis. Publication bias was explored using Egger’s linear regression test [37]. The asymmetry of the funnel plots suggests the presence of small study effects. 

### 2.5. Statistical Analysis

We conducted a proportional meta-analysis using the “metaprop” function, included in the “meta” and “metafor” packages in the R statistical software, version 4.2.3 (R Foundation for Statistical Computing), for efficacy and safety outcomes. Treatment effects for binary endpoints were grouped, and odds ratios (ORs) with 95% confidence intervals were used. Heterogeneity was assessed using the I² statistic; *p*-values of less than 0.05 and I² > 25% were considered significant for heterogeneity. We used a fixed effects model for outcomes with I² < 25% (low heterogeneity). For grouped results with high heterogeneity, the DerSimonian and Laird random effects model was used.

Subgroups of interest included prior therapy and pRCC subtype. Thus, we sought to answer the following question: which of these MET kinase inhibitors are the most suitable for treating pRCC? 

## 3. Results

### 3.1. Study Selection and Characteristics

Nine studies were included: three clinical trials and six observational cohort studies. The search strategy identified a total of 2194 records. After removing duplicated titles, 1373 studies were evaluated through their titles and abstracts according to eligibility criteria, and of these, 19 were eligible for full-text reading. Then, ten were excluded due to overlapping populations or duplicated articles, and nine were included for quantitative analysis, as described in Figure 1. 

We have summarized in Table 1 all the MET inhibitors used in the studies of our meta-analysis, including cabozantinib, foretinib, crizotinib, tivantinib, and savolitinib.

We included nine studies with 504 patients in the meta-analysis. All patients who had advanced (stage IV) pRCC received MET inhibitor therapy. The overall population consisted of 76.2% males and 82.3% were of white ethnicity. The most prevalent histological type was type 2, accounting for 40.2% of the cases, and 31.4% of patients were MET-driven positive (MET+). Baseline characteristics of studies are described in Table 2.

### 3.2. Objective Response Rate

The ORR was 36.4% (95% confidence interval (CI): 10.2; 62.5) for MET+ patients, and only 0.4% (95%CI: 0.0–3.3) for MET-independent patients with a significant difference between groups (*p*-value < 0.01); and an overall population ORR of 20.6% (95%CI: 0.5–40.7), as described in Figure 2. In the subgroup analysis of tumor response according to the treatment administered, cabozantinib, savolitinib, and other therapies, we found a significant difference among subgroups (*p* < 0.01). Patients on cabozantinib had an ORR of 26.1% (95%CI: 18.2–34.0), on savolitinib had an ORR of 15.4% (95%CI: 2.5–28.2), and on other therapies, an ORR of 4.4% (95%CI: 0.0–10.7); with an overall ORR of 16.48% (95%CI: 8.62–24.35) and a total of 75 events out of 504 patients included in the study. The analysis revealed a high level of heterogeneity, with an I^2^ value of 84%. 

### 3.3. Disease Control Rate

The overall population’s disease control rate (DCR) was 69.6% (95%CI: 51.6–87.6) with 249 events out of 413 patients. Figure 3 shows a high level of heterogeneity, with an I^2^ value of 97%.

### 3.4. Progression-Free Survival

Progression-free survival rate at 12 months was 15.1% (95%CI: 9.8–20.3), with 29 events out of 172 patients included in the analyses, described in Figure 4. It had a high level of heterogeneity, with an I^2^ of 52%.

### 3.5. Survival Rate

Utilizing a random effects model for overall survival (OS), patients alive at 12 and 24 months were, respectively, 43.2% (95%CI: 22.5–64.0) and 10.0% (95%CI: 0.0–30.0). At 12 months, we observed 65 events out of 172 patients. The analysis exhibited a high level of heterogeneity, with an I^2^ value of 83%. For OS analysis at 24 months, nine events were observed out of 122 patients included in the study. It showed considerable heterogeneity, with an I^2^ value of 83%. 

### 3.6. Adverse Events

Regarding adverse events (AEs), 95.5% (95%CI: 91.0–99.9) of patients experienced an AE of any grade, as shown in Figure 5. Furthermore, 43.7% (95%CI: 37.1–50.3) of patients had a grade ≥3 AE, indicating severe adverse events occurring in a significant proportion of the population. A random effects model was employed for any grade AE, with a total of 253 AE events out of 266 patients included in the study. The analysis demonstrated a high level of heterogeneity, with an I^2^ value of 65%. The analysis of adverse events (AEs) of grade ≥ 3 in the overall population revealed 160 AEs out of 357 patients. There was a high level of heterogeneity, with an I^2^ value of 27%.

### 3.7. Quality Assessment

Overall, all included studies were at medium to low risk of bias. The SWOG, Sumanta et al. [44], and SAVOIR trials were analyzed with the RoB2 tool, all of which are low-risk studies. For the observational studies, risk of bias assessment through ROBINS-I showed a low-risk in the Choueiri et al. 2013 [38], Choueiri et al. 2017 [39], and CALYPSO studies [46], and a moderate risk for the CREATE, Chanzá et al. 2019 [42], and Tachibana et al. 2022 [45] studies. The tables using RoB2 and ROBINS-I tools are available in Appendix A.

We summarized the sensitive leave-one-out analyses for ORR (Appendix A), DCR (Appendix A), PFS 12 months (Appendix A), OS 12 months (Appendix A), and OS 24 months (Appendix A). The funnel plot distribution in Appendix A represents the ORR analysis (ORR: z value = −6.0860 and *p* < 0.0001), indicating a significant asymmetry. In Appendix A, we present the funnel plot for DCR (z value of 0.2001, *p* = 0.8414), without significant asymmetry, and PFS in 12 months with significant asymmetry (z value of 2.2030, *p* = 0.0276) in Appendix A. Both 12- (Appendix A) and 24-month (Appendix A) OS rate funnel plots showed a significant asymmetry (12 months, z = 2.5624, *p* = 0.0104; and 24 months, z = 3.2096, *p* = 0.0013). In adverse event analysis, the any-grade funnel plot revealed significant asymmetry (z = −2.2357, *p* = 0.0254) (Appendix A), while the grade greater or equal to 3 was not significantly asymmetric (z = −1.4191, *p* = 0.1559) (Appendix A).

## 4. Discussion

In this systematic review and meta-analysis, we assessed the efficacy and safety of MET kinase inhibitors in patients with advanced papillary renal-cell carcinoma. The most relevant findings included ORR for the entire population of 16.5%, and the difference in ORR between MET+ and MET− patient groups emerged as statistically significant (*p* = 0.01). The ORR was found to be 35% for MET+ patients, but only 1% for MET-independent patients. These findings underscore the potential for MET inhibitors to exert a more pronounced therapeutic effect in MET-driven tumors. 

Kidney cancer is the ninth most common cancer worldwide [2]. Renal-cell carcinoma (RCC) accounts for more than 90% of kidney tumors and exhibits diverse subtypes and a wide range of clinical outcomes [47,48]. Among non-clear-cell RCC, the pRCC, represents 15% of all RCCs [49,50,51,52]. The MET proto-oncogene has been found to have mutations and copy number changes based on the genomic and molecular characterization of pRCC. pRCC type 2 was the most common histological subtype in the present meta-analysis, accounting for 40.2% of the cases, and 31.4% of the patients had a MET-driven (MET+) status. Although patients with type 1 disease are more likely to have MET alterations than patients with type 2 [53], current evidence indicates that the type 2 disease may also be characterized by molecular changes in the MET oncogene and/or activated MET pathway signaling [50,54]. Thus, MET inhibitors are a promising therapy for MET-driven PRCC, as analyzed in the present single-arm study.

In preclinical models, drugs like crizotinib, savolitinib, and cabozantinib have shown potent MET inhibitory properties [55,56,57]. When stratified by MET inhibitor therapies, savolitinib had an ORR of 15.4%, cabozantinib had a 26.1% ORR, and other treatments had an ORR of 4.4%. 

The c-MET proto-oncogene is located in band 7q21-q31 of chromosome 7 and is responsible for the fusion of the translocated promoter region (TPR)-Met [20]. To produce the Met protein, the post-Golgi compartment is responsible for the glycosylation and proteolytic cleavage of its precursor: a single chain with a molecular weight of 170 kDa [54]. Its activation is responsible for promoting the Ras/Raf/MEK/MAPK, Ras/PAK pathways, SRC, FAK, JAK/STAT, Wnt/β-catenin, and PI3K/Akt/mTOR signaling pathways [58]. Aberrant c-MET signaling has been previously described in association with tumors, occurring due to amplification, rearrangement, and autocrine and paracrine stimulation, thus leading to increased motility, migration, angiogenesis, and metastasis [56].

Regarding pharmacodynamics, small molecules inhibit a variety of targets [44,57]. VEGF expression in PRCC is comparable to, if not higher, than in clear-cell disease, even though changes in VHL and the resulting increases in VEGF have historically been associated with clear-cell disease [59]. Due to the possibility that PRCC is regulated by both VEGF and MET signaling pathways, dual VEGF/MET inhibitors like cabozantinib may be more effective at targeting PRCC, as supported by our meta-analysis.

Additionally, VEGF, MET, and AXL are all targets for foretinib and cabozantinib. AXL, a tyrosine kinase receptor overexpressed in both clear-cell and papillary RCC histology, may be a key therapeutic target [60]. This contrasts with savolitinib, a more selective MET inhibitor [57], and may account for cabozantinib’s better ORR results. Despite the fact that both foretinib and crizotinib have significant polypharmacology with kinases other than MET, further research is needed to determine how well they will perform in PRCC.

The majority of our study population was white (82.3%). These data contrast with the prevalence of pRCC, which is more common in people of African or Afro-Caribbean descent, while ccRCC is more common in Caucasians [12,61,62,63]. These discrepancies may be explained by the low representation of black participants in clinical trials. The study by Bebi et al. evaluated the racial composition of 8 million patients from 27,000 clinical trials conducted in the USA (oncology n = 118,194, cardiovascular n = 12,281, central nervous system n = 35,533) and found that only 8.5% of oncology trial patients % were black [64].

Any-grade AE was reported in 95.5% of patients, and grade-3 or higher AE occurred in 43.7% of patients. Savolitinib’s higher MET selectivity may be responsible for its lower toxicity than sunitinib, a multikinase inhibitor, as the SAVOIR trial showed sunitinib to have a lower tolerability than the MET inhibitor [43]. This was also noted in the Sumanta et al. trial, where more selective MET inhibitors savolitinib and crizotinib, respectively, had 39% and 37% of patients with grade 3 AE [44]. Additionally, the proportions of grade-3 AEs for cabozantinib and sunitinib were similar, at 74% and 69%, respectively [44].

For quality assessment, funnel plot analyses revealed a significant asymmetry in all the following outcomes: ORR, PFS rate at 12 months, overall survival rate at 12 and 24 months, and any-grade adverse events. This indicates a significant probability of publication bias, which may be explained by MET inhibitors’ performance in the overall population, without selection by MET status, being poorer than in MET+ patients. Thus, there might be unpublished studies due to negative results in a population, including both MET+ and MET- patients. 

The main limitation of our analysis was the insufficient number of randomized controlled trials (RCTs), so we also included observational studies. Among RCTs, a limiting factor was the reduced number of patients, which made it challenging to analyze separately each intervention and their performance on MET-driven patients. Furthermore, the results were significantly better than in MET-independent patients. Another limitation is that our study was primarily composed of white people, and the extrapolation of our data to other populations should be made with caution. In the future, larger studies should be conducted in MET-driven pRCC patients with the most successful MET inhibitor therapies, i.e., cabozantinib, an example of good performance, with a better ORR when compared to other MET inhibitors. 

## 5. Conclusions

This is the first systematic review and meta-analysis to bring together the available data on the efficacy and safety of MET inhibitors in patients with advanced papillary renal-cell carcinoma. Our results support that MET inhibitors have an antitumor effect and can benefit these patients, particularly those with MET-driven tumors.

## Figures and Tables

**Figure 1 ijms-24-17582-f001:**
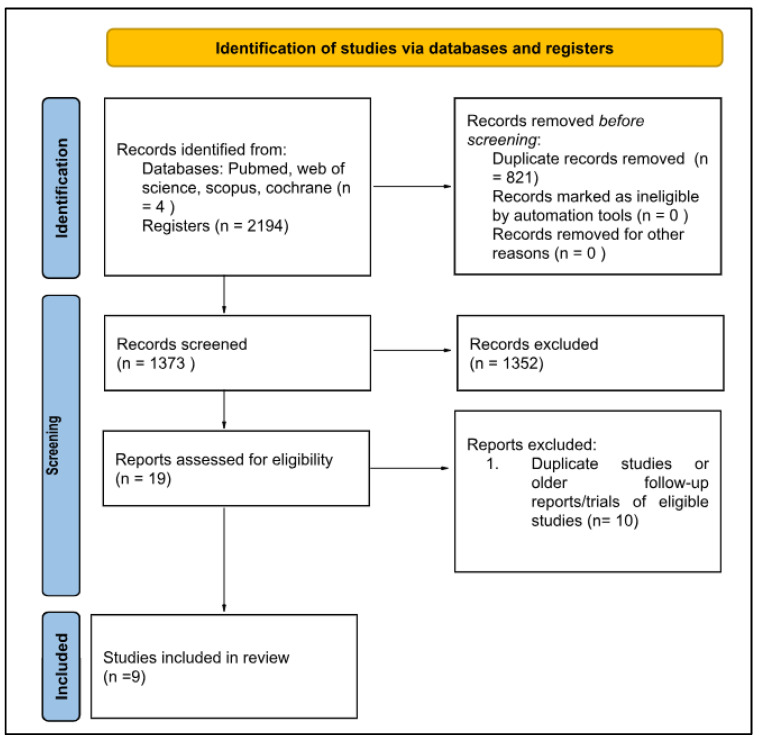
Flow diagram of the study selection.

**Figure 2 ijms-24-17582-f002:**
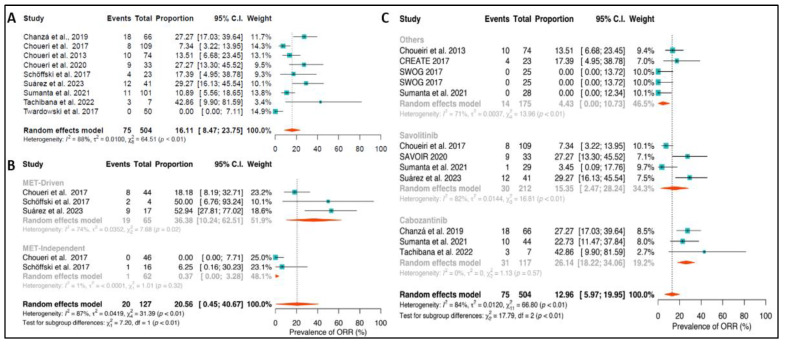
(**A**) Objective response rate (ORR) in the overall population, ORR stratified by MET status (**B**) and ORR stratified by cabozantinib, savolitinib, or other MET inhibitor therapies (**C**) [38,39,40,41,42,43,44,45,46].

**Figure 3 ijms-24-17582-f003:**
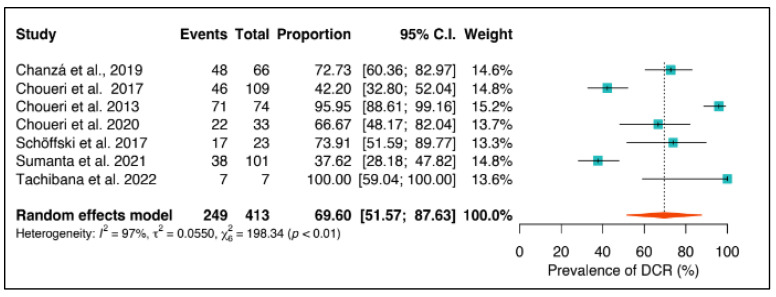
Disease control rate in the overall population [38,39,40,42,43,44,45].

**Figure 4 ijms-24-17582-f004:**
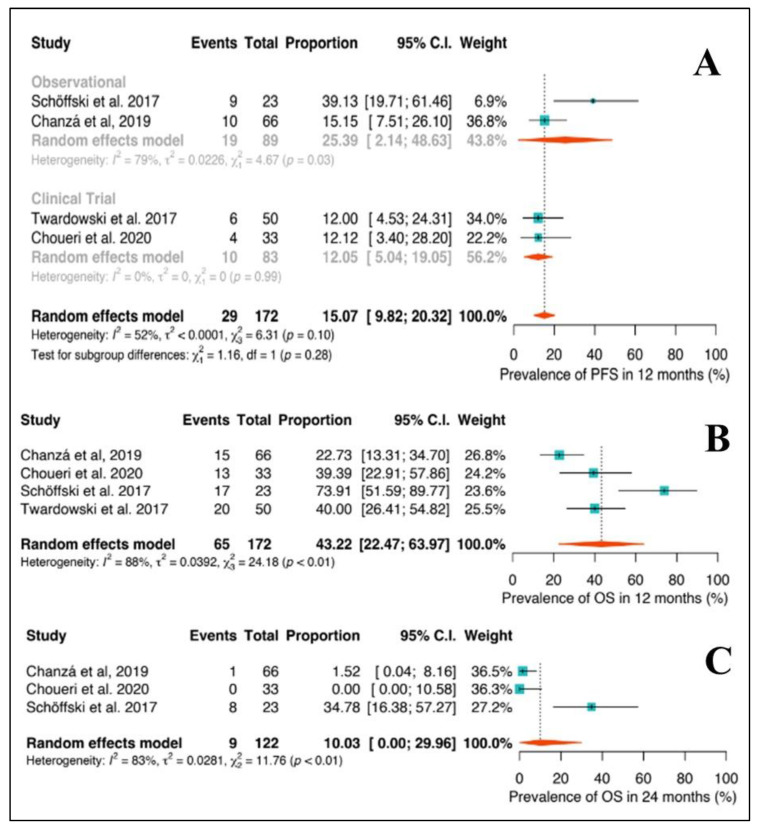
Progression-free survival at 12 months stratified by study design (**A**). Overall survival rate at 12 months in the overall population (**B**). Overall survival rate at 24 months in the overall population (**C**) [40,41,42,43].

**Figure 5 ijms-24-17582-f005:**
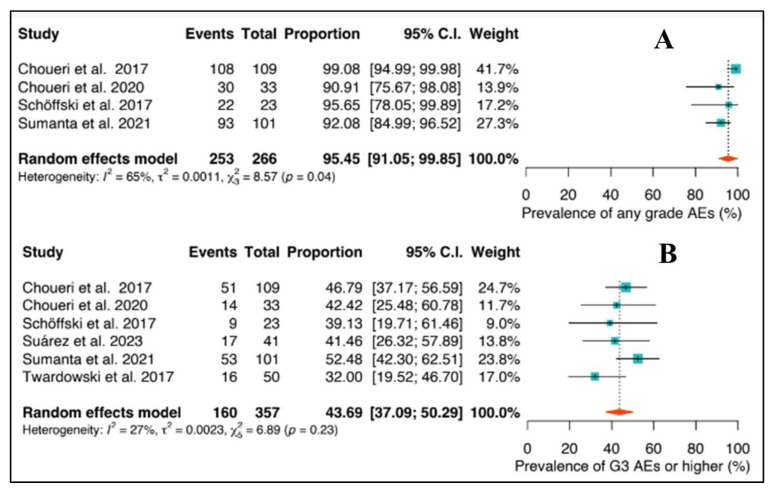
(**A**) Adverse events (any grade) and (**B**) adverse events (grade ≥ 3) in the overall population [39,40,41,43,44,46].

**Table 1 ijms-24-17582-t001:** Characterization of MET inhibitors included in the meta-analysis.

Compound	Molecular Formula	Mechanism of Action
**Non-selective small molecule inhibitor**		
Cabozantinib	C_28_H_24_FN_3_O_5_	Inhibitor of MET, VEGFR2, and RET kinases
Crizotinib	C_21_H_22_C_l2_FN_5_O	Inhibitor of ALK, HGFR, c-Met, and RON
Foretinib	C_34_H_34_F_2_N_4_O_6_	ATP-competitive inhibitor of HGFR, VEGFR, MET, and KDR
**Selective small molecule inhibitor**		
Tivantinib	C_23_H_19_N_3_O_2_	Inhibitor of MET, EGFR, InsR, PDGFRα, or FGFR1/4.
Savolitinib	C_17_H_15_N_9_	Inhibitor of MET

**Table 2 ijms-24-17582-t002:** Study characteristics.

Study	Design	Follow-Up (mo)	No. of Patients	Age †, yr	Male Sex	Performance Status *	Intervention	Race	Dosing Schedule (P.O.)
Choueiri 2013 [38]	Phase II Prospective Cohort	N/A	74	57	59 (80%)	0: 54 (73%) I–II: 20 (27%)	Foretinib; 2w	64 (86.5%)white 10 (13.5%) nonwhite *	CA: 240 mg days 1–5 CB: 80 mg daily
Choueiri 2017 [39]	Phase II Prospective cohort	N/A	109	64	78 (72%)	0: 51 (47%) I: 58 (53%)	Savolitinib; 3w	96 (88%)white 13 (12%) nonwhite *	600 mg daily
CREATE 2017 [40]	Prospective cohort	26.8	23	62.5	20 (87%)	0: 13 (56.5%) 1: 10 (43.5%)	Crizotinib; 3w	N/A	250 mg BID
SWOG 2017 [41]	Phase II RCT (Arm A)	N/A	25	62.1	19 (76%)	0: 12 (48%) I-II:13 (52%)	Tivantinib; 4w	19 (76%) white 6 (24%) nonwhite *	360 mg BID
SWOG 2017 [41]	Phase II RCT (Arm B)	N/A	25	63.6	15 (60%)	0: 9 (36%) I-II: 16 (64%)	Tivantinib + Erlotinib; 4w	19 (76%) white 6 (24%) nonwhite *	360 mg BID
Chanzá 2019 [42]	Retrospective cohort	11	66	N/A	N/A	0: 13 (56.5%) I: 10 (43.5%)	Cabozantinib	N/A	93 (83%) pts: 60 mg daily 19 (17%) pts: started 40 mg daily; 2 pts: increased to 60 mg daily
SAVOIR 2020 [43]	Phase III RCT	12	33	60	29 (88%)	0: 26 (78%) I: 7 (21%)	Savolitinib; 6w (4 tx, 2 none)	29 (88%) white 4 (12%) nonwhite *	600 mg daily
Sumanta 2021 [44]	Phase II RCT (Arm A)	Up to 36	44	65	36 (82%)	0: 29 (66%) I: 15 (34%)	Cabozantinib; 6w (4 tx, 2 none)	32 (73%) white 12 (27%) nonwhite *	60 mg daily
Sumanta 2021 [44]	Phase II RCT (Arm B)	Up to 36	28	68	22 (79%)	0: 18 (64%) I: 10 (36%)	Crizotinib; 6w (4 tx, 2 none)	22 (79%) white 6 (22%) nonwhite *	250 mg BID
Sumanta 2021 [44]	Phase II RCT (Arm C)	Up to 36	29	67	19 (66%)	0: 15 (52%) I: 14 (48%)	Savolitinib; 6w (4 tx, 2 none)	21 (72%) white 8 (26%) nonwhite *	600 mg daily
Tachibana 2022 [45]	Retrospective cohort	7.1	7	N/A	N/A	N/A	Cabozantinib	N/A	N/A
CALYPSO 2023 [46]	Phase II Prospective Cohort	N/A	41	N/A	N/A	N/A	Savolitinib + Durvalumab	N/A	600 mg daily

†, median; HF, histological feature at initial diagnosis; Type 1 or 2: pRCC type 1 or 2; Mo, months; N/A, Not available; Non-white *, all groups included in non-white; Others, all other groups included; BID, twice a day; Pts, patients; d, days; w, weeks; CA, cohort A; CB, cohort B.

## Data Availability

Data is contained within the article and Appendix A.

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
