# Peer review of "Mesenchymal–Epithelial Transition Kinase Inhibitor Therapy in Patients with Advanced Papillary Renal-Cell Carcinoma: A Systematic Review and Meta-Analysis"

_ijms, 2023, doi:10.3390/ijms242417582_

Round 1

Reviewer 1 Report

Comments and Suggestions for Authors

Dear authors,

This is  avery nice paper. However some points need to be clrified:

*) In table 1 list of all  studies considered are included. However I could not find some of he studies in this table, an they are more than 9

**= Forest plots  would be clarer if the vertical line was placed on 0.

***) PLwase make sure that the same cases are not included in different studies

****) In all studies white population is higher. Authors shoould comment on this and try to related with other clinical variables. Maybe they should add some limitationes pf the study regarding this point

Author Response

November 24, 2023

Re: MET Kinase Inhibitor Therapy In Patients With Advanced Papillary Renal Cell Carcinoma: A Systematic Review And Meta-Analysis

Francisco Cezar Aquino de Moraes, Maysa Vilbert, Vinícius Freire Costa Alves, Gustavo de Oliveira Almeida, Jonathan N. Priantti, Thiago Madeira, Carlos Stecca, Marianne Rodrigues Fernandes and Ney Pereira Carneiro dos Santos

Dear Reviewer 1,

Editorial Board Member of International Journal of Molecular Science,

We are very appreciative of the careful reading which you gave our manuscript.  We meticulously addressed each of the points and questions raised and performed the corrections or justifications accordingly. In the discussion that follows we have indicated our direct responses to the questions raised by your thorough review. In addition, we have highlighted in yellow the manuscript changes made according to your suggestions. The manuscript is significantly improved because of your constructive criticisms and hope that you will find the revised version of this manuscript acceptable.

RESPONSE TO REVIEWER  #1

Dear authors,

This is  avery nice paper. However some points need to be clrified:

  • In table 1 list of all  studies considered are included. However I could not find some of he studies in this table, an they are more than 9

In response: We thank the reviewer for the thoughtful consideration of our manuscript. In our meta-analysis, we included 9 studies for the qualitative and quantitative analysis. However, Sumanta 2021 [44] and SWOG 2017 [41] are randomized clinical trials with three and two arms, respectively. Each arm has different drugs and dosing schedules. Therefore, we chose to describe each arm individually in a row as we further included them separately in our pooled analysis. Because of this, we have 12 rows for 9 studies in Table 1. To make it clear to our readers, we added this information in the table, in the column "Design" as: Arm A, Arm B, and Arm C. The nine studies are: Choueiri 2013 [38]; Choueiri 2017 [39]; CREATE 2017 [40]; SWOG 2017 [41]; Chanzá 2019 [42]; SAVOIR 2020 [43]; Sumanta 2021 [44]; Tachibana 2022 [45]; CALYPSO 2023 [46].  

**= Forest plots would be clarer if the vertical line was placed on 0.

In response: Thank you for your suggestion. We performed a proportional meta-analysis or single-arm meta-analysis, pooling the studies to evaluate the overall proportion (frequency) of our outcomes. Then, in this case, the vertical line represents the overall proportion of each analysis; the individual studies are distributed around the vertical line according to their contribution to the overall proportion; and the diamond represents the overall proportion’s 95% confidence interval. The vertical line would be placed on zero only if the overall proportion of the forest plot was zero. Otherwise, it could be anywhere between 0 and 100%. Because of this, we could not make the suggested change, as, in this case, the final result (the overall proportion) defines the position of the vertical line.

  • ***) Please make sure that the same cases are not included in different studies

In response: We really appreciate your comment. We carefully reviewed it all again, and the studies do not have an overlapping population; they have different patients enrolled in various centers and diverse recruitment periods. We have also rechecked all our analyses and confirmed no overlapping population.

  • ****) In all studies white population is higher. Authors shoould comment on this and try to related with other clinical variables. Maybe they should add some limitationes pf the study regarding this point

In response: Thank you so much for your comment. This is a very important topic. We added a paragraph about it in the discussion section. We discussed the representativeness of different ethnicities in clinical trials and its implications for the results obtained. In addition, We included it in the study's limitations, as you suggested.

Reviewer 2 Report

Comments and Suggestions for Authors

Congratulations for you work, you focused on a focal point in the evolution of personalized and tailored therapies. The landscape of treatment of both classical clear cell RCC and non clear Cell RCC are dramatically changing in the last years and mutiple protocolos are arising focusing on immunotherapy and association and timing.

Iacovelli R, Cannella MA, Ciccarese C, Astore S, Foschi N, Palermo G, Tortora G. 2021 ASCO genitourinary cancers symposium: a focus on renal cell carcinoma. Expert Rev Anticancer Ther. 2021 Nov;21(11):1203-1206. doi: 10.1080/14737140.2021.1976147. Epub 2021 Sep 9. PMID: 34482771.

IReading this kind of paper lead every one to consider 8 years as 50 years.Courthod G, Tucci M, Di Maio M, Scagliotti GV. Papillary renal cell carcinoma: A review of the current therapeutic landscape. Crit Rev Oncol Hematol. 2015 Oct;96(1):100-12. doi: 10.1016/j.critrevonc.2015.05.008. Epub 2015 May 27. PMID: 26052049.

Your work underline the need of more tailored approaches and the urgency of well designed RCTs.

Author Response

November 24, 2023

Re: MET Kinase Inhibitor Therapy In Patients With Advanced Papillary Renal Cell Carcinoma: A Systematic Review And Meta-Analysis

Francisco Cezar Aquino de Moraes, Maysa Vilbert, Vinícius Freire Costa Alves, Gustavo de Oliveira Almeida, Jonathan N. Priantti, Thiago Madeira, Carlos Stecca, Marianne Rodrigues Fernandes and Ney Pereira Carneiro dos Santos

Dear Reviewer 2,

Editorial Board Member of International Journal of Molecular Science,

We are very appreciative of the careful reading which you gave our manuscript.  We meticulously addressed each of the points and questions raised and performed the corrections or justifications accordingly. In the discussion that follows we have indicated our direct responses to the questions raised by your thorough review. In addition, we have highlighted in yellow the manuscript changes made according to your suggestions. The manuscript is significantly improved because of your constructive criticisms and hope that you will find the revised version of this manuscript acceptable.

RESPONSE TO REVIEWER #2

Congratulations for you work, you focused on a focal point in the evolution of personalized and tailored therapies. The landscape of treatment of both classical clear cell RCC and non clear Cell RCC are dramatically changing in the last years and mutiple protocolos are arising focusing on immunotherapy and association and timing.

Iacovelli R, Cannella MA, Ciccarese C, Astore S, Foschi N, Palermo G, Tortora G. 2021 ASCO genitourinary cancers symposium: a focus on renal cell carcinoma. Expert Rev Anticancer Ther. 2021 Nov;21(11):1203-1206. doi: 10.1080/14737140.2021.1976147. Epub 2021 Sep 9. PMID: 34482771.

IReading this kind of paper lead every one to consider 8 years as 50 years.Courthod G, Tucci M, Di Maio M, Scagliotti GV. Papillary renal cell carcinoma: A review of the current therapeutic landscape. Crit Rev Oncol Hematol. 2015 Oct;96(1):100-12. doi: 10.1016/j.critrevonc.2015.05.008. Epub 2015 May 27. PMID: 26052049.

Your work underline the need of more tailored approaches and the urgency of well designed RCTs.

In response: Thank you very much for your valuable comments. We have added a topic in the introduction that explores emerging therapies, particularly immune checkpoint inhibitors, which have been the focus in recent years.

Thank you for your suggestions for references. We have added them to the manuscript.

In the Discussion section, we have added, as highlighted, the limitations of this study, raising the urgent need to conduct RCTs in this scenario.

Reviewer 3 Report

Comments and Suggestions for Authors

MET is a tyrosine kinase receptor frequently altered in type 1 papillary renal cell carcinoma (PRCC). Recently, it has been discovered that type 2 PRCC can also be MET-driven, although it is less common than in type 1 PRCC.

Numerous small molecule inhibitors have been developed to selectively target MET, such as savolutinib and tivatinib. Additionally, multiple tyrosine kinase inhibitors (mTKIs), like crizotinib, cabozantinib, and foretinib, have been studied in clinical trials for their ability to target MET among other tyrosine kinases. Some of these inhibitors, such as cabozantinib, have been approved for the treatment of renal cell carcinoma (RCC), while others, like crizotinib, have been used to treat other histological types of tumors with MET mutations. Monoclonal antibodies that target MET or block the interaction between HGF and MET have also been developed.

The authors present a meta-analysis of MET inhibition in advanced PRCC, which is particularly intriguing because PRCC accounts for 15-20% of RCC cases, and most clinical research has primarily focused on the clear cell type RCC. Type 2 PRCC, with its dismal prognosis and general resistance to modern molecular targeting drugs, represents an understudied area.

The study's flow diagram, depicted in Figure 1, and the research methods employed are relevant to the study's objectives.

The meta-analysis revealed an overall response rate of 36% in MET-driven patients. Although adverse events of any grade were observed in 96% of the patients, severe adverse events were found in less than half of the patients. The authors conclude that MET inhibitors can be a viable treatment option for MET-driven PRCC patients.

Before the manuscript can be accepted, several points should be addressed:

1. The authors should revise the introduction to make it more lucid, with greater attention to information on types 1 and 2 PRCC, including genetic alterations. Information on hereditary RCC is irrelevant to the present meta-analysis and can be omitted.

2. The mechanism of MET activation should be thoroughly discussed, encompassing aspects like overexpression, gene amplification, point mutations, and so forth.

3. The authors should provide a more comprehensive summary of MET inhibitors in a separate table, clearly indicating whether each is a small molecule or an antibody, and specifying whether it is a specific MET inhibitor or an mTKI that can inhibit MET among other targets.

Author Response

November 24, 2023

Re: MET Kinase Inhibitor Therapy In Patients With Advanced Papillary Renal Cell Carcinoma: A Systematic Review And Meta-Analysis

Francisco Cezar Aquino de Moraes, Maysa Vilbert, Vinícius Freire Costa Alves, Gustavo de Oliveira Almeida, Jonathan N. Priantti, Thiago Madeira, Carlos Stecca, Marianne Rodrigues Fernandes and Ney Pereira Carneiro dos Santos

Dear Reviewer 3,

Editorial Board Member of International Journal of Molecular Science,

We are very appreciative of the careful reading which you gave our manuscript.  We meticulously addressed each of the points and questions raised and performed the corrections or justifications accordingly. In the discussion that follows we have indicated our direct responses to the questions raised by your thorough review. In addition, we have highlighted in yellow the manuscript changes made according to your suggestions. The manuscript is significantly improved because of your constructive criticisms and hope that you will find the revised version of this manuscript acceptable.

RESPONSE TO REVIEWER #3

MET is a tyrosine kinase receptor frequently altered in type 1 papillary renal cell carcinoma (PRCC). Recently, it has been discovered that type 2 PRCC can also be MET-driven, although it is less common than in type 1 PRCC.

Numerous small molecule inhibitors have been developed to selectively target MET, such as savolutinib and tivatinib. Additionally, multiple tyrosine kinase inhibitors (mTKIs), like crizotinib, cabozantinib, and foretinib, have been studied in clinical trials for their ability to target MET among other tyrosine kinases. Some of these inhibitors, such as cabozantinib, have been approved for the treatment of renal cell carcinoma (RCC), while others, like crizotinib, have been used to treat other histological types of tumors with MET mutations. Monoclonal antibodies that target MET or block the interaction between HGF and MET have also been developed.

The authors present a meta-analysis of MET inhibition in advanced PRCC, which is particularly intriguing because PRCC accounts for 15-20% of RCC cases, and most clinical research has primarily focused on the clear cell type RCC. Type 2 PRCC, with its dismal prognosis and general resistance to modern molecular targeting drugs, represents an understudied area.

The study's flow diagram, depicted in Figure 1, and the research methods employed are relevant to the study's objectives.

The meta-analysis revealed an overall response rate of 36% in MET-driven patients. Although adverse events of any grade were observed in 96% of the patients, severe adverse events were found in less than half of the patients. The authors conclude that MET inhibitors can be a viable treatment option for MET-driven PRCC patients.

Before the manuscript can be accepted, several points should be addressed:

  • The authors should revise the introduction to make it more lucid, with greater attention to information on types 1 and 2 PRCC, including genetic alterations. Information on hereditary RCC is irrelevant to the present meta-analysis and can be omitted.

In response: Thank you for your suggestion. We reorganized the introduction to make it more lucid, highlighting the molecular alterations on pRCC types 1 and 2. We have also omitted information on germline or hereditary alterations. Thank you.

  • The mechanism of MET activation should be thoroughly discussed, encompassing aspects like overexpression, gene amplification, point mutations, and so forth.

In response: We thank the reviewer for this suggestion. We made the suggested adjustments and added a topic on the mechanisms of MET activation, including cell signaling.

  • The authors should provide a more comprehensive summary of MET inhibitors in a separate table, clearly indicating whether each is a small molecule or an antibody, and specifying whether it is a specific MET inhibitor or an mTKI that can inhibit MET among other targets.

In response: Thank you for this suggestion. We added to the results table 1 summarizing the MET inhibitors included in this study, including the molecular formula and mechanism of action of each drug.
